# Quantifying the impact of an invasive hornet on *Bombus terrestris* colonies

Thomas A. O'Shea-Wheller [1✉], Robin J. Curtis[1], Peter J. Kennedy [1], Ellen K. J. Groom[1], Juliette Poidatz[1,2], David S. Raffle [1], Sandra V. Rojas-Nossa[3], Carolina Bartolomé [4], Damián Dasilva-Martins[4], Xulio Maside[4], Salustiano Mato[3] & Juliet L. Osborne [1]

The invasive hornet *Vespa velutina nigrithorax* is considered a proliferating threat to pollinators in Europe and Asia. While the impact of this species on managed honey bees is well-documented, effects upon other pollinator populations remain poorly understood. Nonetheless, dietary analyses indicate that the hornets consume a diversity of prey, fuelling concerns for at-risk taxa. Here, we quantify the impact of *V. velutina* upon standardised commercially-reared colonies of the European bumblebee, *Bombus terrestris terrestris*. Using a landscape-scale experimental design, we deploy colonies across a gradient of local *V. velutina* densities, utilising automated tracking to non-invasively observe bee and hornet behaviour, and quantify subsequent effects upon colony outcomes. Our results demonstrate that hornets frequently hunt at *B. terrestris* colonies, being preferentially attracted to those with high foraging traffic, and engaging in repeated—yet entirely unsuccessful—predation attempts at nest entrances. Notably however, we show that *B. terrestris* colony weights are negatively associated with local *V. velutina* densities, indicating potential indirect effects upon colony growth. Taken together, these findings provide the first empirical insight into impacts on bumblebees at the colony level, and inform future mitigation efforts for wild and managed pollinators.

[1] Environment and Sustainability Institute, University of Exeter, Penryn, Cornwall TR109FE, United Kingdom. [2] UMR PVBMT, Peuplements Végétaux et Bioagresseurs en Milieu Tropical, CIRAD, 97410 Saint Pierre, La Réunion, France. [3] Department of Ecology and Animal Biology, Faculty of Sciences, University of Vigo, 36310 Vigo, Pontevedra, Spain. [4] Grupo de Medicina Xenómica, CIMUS, Universidade de Santiago de Compostela, 15781 Santiago de Compostela, A Coruña, Spain. ✉email: t.a.oshea-wheller@exeter.ac.uk

The maintenance of pollination services is a global concern, as current estimates indicate that ~87.5% of all terrestrial angiosperms rely upon animals for reproduction[1]. When considering agriculture, this reliance is further intensified to a relatively constrained group of insect taxa, with bees alone visiting >90% of leading crop types[2,3]. As such, reductions in pollinator health constitute a threat to both biodiversity and food security at the landscape-scale, with consequent efforts being made to conserve a variety of at-risk species[4,5]. Among these, wild bees are of specific concern, as they do not benefit from anthropogenic management[6] and comparatively little data exists to monitor natural population changes[7,8]. While substantial efforts have been made to characterise the drivers of pollinator declines—with habitat loss emerging as a primary threat[2,7,9]—there remains significant uncertainty regarding a plethora of additional prospective factors, indicating the need for further research[7,9,10].

Interestingly, in assessments of such factors, the impact of invasive species is frequently ranked below other threats to pollinators, ostensibly due to a paucity of data[2,7]. While caution is warranted, it is important to note that a deficiency of data is not tantamount to an absence of effect, and instead underlines the need for additional investigation[10,11]. This point is especially pertinent when considering that other drivers of pollinator decline with comparatively little quantitative evidence appear to benefit substantially from the 'precautionary principle', combined with anthropocentric bias[7,10,12]. Nevertheless, there is a growing body of evidence that invasive organisms exert profound, though often poorly-visible, impacts upon wild pollinators and pollination services[13–15]. As such, basic empirical data is urgently needed to better identify and mitigate the risks posed by these organisms.

An emerging global threat is that of the hornet *Vespa velutina nigrithorax*—colloquially known as the yellow-legged or Asian hornet—the first insect to be legislatively classified as an invasive alien species of concern in Europe[16]. This social vespid is native to southeast Asia, but has spread rapidly through parts of east Asia and Europe, following accidental introductions in or before 2003 and 2004 respectively[17–19]. The invasion front has expanded at a rate of up to 78 km per year in some regions[18,20], facilitated by an adaptable life-history[21], rapid colony population growth[18], and the ability of mated queens to travel substantial distances in a single day[22]. Control efforts are further hampered by the difficulty of locating nests[20,23], and high potential for additional human-mediated dispersal events[24], as is the case with many other invasive eusocial insects[25–28]. Consequently, efforts to contain the spread of these hornets have had limited success[18,21], while climatic niche modelling predicts their continued expansion into new regions[29,30].

The capacity of *V. velutina* to impact insect pollinators is evident through their highly-visible and intense predation of the honey bee *Apis mellifera*[21]. While these hornets are natural predators of the congeneric *Apis cerana*—which has consequently evolved a suite of defensive behaviours[31]—*A. mellifera* constitutes an evolutionarily naïve prey species, and thus has few effective defence mechanisms, enabling the hornets to predate upon colonies with near-impunity[31–33]. As such, predation pressure can become so severe that foraging is markedly curtailed, leading to the eventual starvation of colonies, with reported mortality rates exceeding 30% in some regions[21].

Although managed honey bees have occupied much of the research focus, preliminary dietary analyses provide strong evidence that *V. velutina* also preys upon a diversity of wild pollinators[34,35]. Indeed, previous work indicates that both hornet predation and foraging activity exert direct effects on plant-pollinator communities[36]. This is concerning, as the broader impact of *V. velutina* upon the survival and reproduction of wild pollinators remains largely unknown[21]. Consequently, given the scale of damage wrought by other invasive Vespidae[37], there is a pressing need for robust data to bridge current knowledge gaps.

One group that may be particularly vulnerable are eusocial bumblebees of the genus *Bombus*. This is because their colony-based lifecycle has the potential to attract *V. velutina* in a similar manner to that of *A. mellifera*; namely by offering a dense aggregation of potential prey at colony entrances. Further, the substantially smaller colony sizes of *Bombus* spp[38]., coupled with ongoing wild population declines in Europe[39–41], arguably place bumblebees at even greater risk than managed honey bees. Currently, the potential impacts of *V. velutina* upon bumblebee colonies remain largely unknown, however, the presence of *Bombus* in the hornets' diet highlights the need for further inquiry[34].

Here, we investigated the effects of *V. velutina* upon colonies of the native European bumblebee *Bombus terrestris terrestris*. Utilising a landscape-scale experimental design, we established standardised *B. terrestris* colonies at field sites across Pontevedra, Spain—a region with substantial heterogeneity in local *V. velutina* densities. Colonies were fitted with automated camera monitoring systems (Fig. 1a, b), allowing for the quantification of hornet and bumblebee activity remotely, along with resultant colony outcomes. Using a combination of detailed behavioural observation, automated tracking of foraging activity, and extensive colony assessments, we then examined fine-scale interactions between the two species, and quantified bumblebee colony survival and reproductive success across a gradient of *V. velutina* densities.

Results demonstrated that foraging *V. velutina* frequently attempted to predate upon *B. terrestris* workers at colony entrances, yet in the present study, all such observed attempts ended in failure. Despite this, bumblebee colony weights showed a negative correlation with hornet densities at sites, suggesting potential indirect effects upon colony growth. Notably, such effects did not appear to influence either colony survival or queen production, indicating a degree of resilience within the timescale assessed. Taken together, this work comprises the first comprehensive colony-level analysis of how *V. velutina* affects a native bumblebee, and provides baseline data with which to better understand the current and future environmental impacts of the hornet.

## Results

**Factors influencing hornet counts at sites**. To assess the impact of *V. velutina* on *B. terrestris* colonies in a representative environment, we established 12 field sites from an initial group of 15 locations across the province of Pontevedra, Spain (Fig. S1a). As we wished to select sites encompassing a range of *V. velutina* densities and land cover types (Figs. S1b, c, S2, and Table S1), we first characterised relative hornet abundance at a local scale. This was achieved using two modified VespaCatch (Véto-pharma) traps per site, providing continuous counts of captured hornets every two days for the duration of the study. We then utilised, these data to provide a temporally relevant measure of hornet abundance at each site, and investigate the influence of local climatic and environmental variables on hornet density.

Hornet counts showed substantial spatiotemporal variability across sites (Fig. 2a), being significantly greater with increased minimum temperature (GLMM, effect of minimum temperature: $F_{1,187} = 90.478$, $P < 0.001$; Fig. 2b), decreased minimum relative humidity (GLMM, effect of minimum relative humidity: $F_{1,225} = 14.170$, $P < 0.001$; Fig. 2c), and a lower percentage of water cover (GLMM, effect of water bodies: $F_{1,13} = 8.988$, $P = 0.011$). Additionally, counts of trapped hornets peaked in

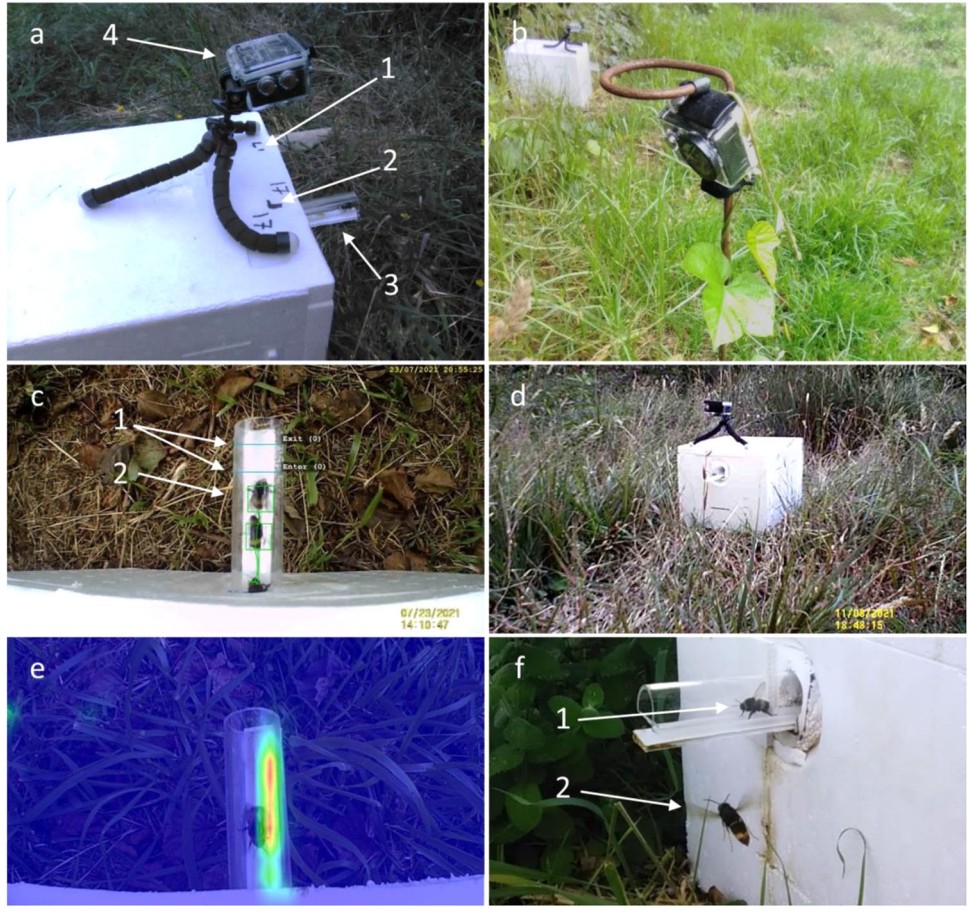

**Fig. 1 Automated monitoring setup. a** Experimental colony setup. Arrows indicate (1) rangefinder markings; (2) frame calibration markings; (3) observation entrance; and (4) entrance camera. **b** External camera set up 1 m from colony. **c** Typical view from the entrance camera demonstrating automated tracking of *B. terrestris* workers. Arrows indicate (1) digital entrance and exit counters; (2) AI-assisted tracking locks (boxes) and trajectories (lines) of individual bumblebees. **d** View from the external camera showing the colony exterior and surroundings. **e** Heatmap detailing the time spent by bumblebees traversing the observation entrance (shorter, blue; longer, red), confirming the optimum region for tracking. **f** A typical interaction between the two species preceding a predation attempt. Arrows denote (1) a *B. terrestris* forager exiting the colony; (2) a *V. velutina* worker investigating the colony entrance.

early-August, and exhibited a decreasing trend thereafter (GLMM, effect of date: $F_{1,19} = 495.081$, $P < 0.001$; Fig. 2b). The percentage of discontinuous urban fabric at sites had no discernible impact (GLMM, effect of discontinuous urban fabric: $F_{1,10} = 3.356$, $P = 0.099$), however, site ID had a significant random effect (site random effect: $Z = 1.958$, $P = 0.050$). Notably, all other land cover variables had little predictive influence, and thus were excluded from analyses.

**Factors Influencing Hornet Activity at Bumblebee Colonies.** Across the course of the study, during set sampling periods, hornet activity was recorded using an external Dragon Touch Vision 1 camera (Dragon Touch), trained on each colony at a distance of 1 m (Fig. 1b). This enabled observation of the colony exterior and proximate surroundings, including hornet and bee interactions in the immediate vicinity (Fig. 1d, f). Semi-automated quantification of hornet presence and predatory behaviour was then conducted using BORIS (release v. 8.5), and combined with colony growth, survival, and foraging data to investigate subsequent effects. Additionally, 10 bee and 10 hornet foragers were caught and weighed each day, to assess changes in the weight ratio of the two species over time, and determine whether this influenced interaction dynamics.

Hornets were regularly observed hovering near bumblebee colony entrances, and attempting to predate upon departing or returning foragers. Specifically, results demonstrated that hornets were more likely to be present at colonies that were heavier (GLMM, effect of colony weight: $F_{1,200} = 4.285$, $P = 0.040$) and had a higher frequency of foraging (GLMM, effect of total bee foraging frequency: $F_{1,200} = 6.174$, $P = 0.014$; Fig. 3a), independent of colony ID (colony random effect: $Z = 0.568$, $P = 0.570$). In contrast however, hornet presence at colonies was inversely correlated with both temperature (GLMM, effect of site temperature: $F_{1,200} = 6.631$, $P = 0.011$) and relative humidity (GLMM, effect of site relative humidity: $F_{1,200} = 7.649$, $P = 0.006$) at the time of sampling.

Notably, despite a total of 125 recorded predation attempts during video sampling spanning a period of 27 days, hornets had a 100% failure rate when attempting to prey upon bumblebees. The frequency of these attempts was not influenced by the weight ratio of individual hornets to bumblebees (GLMM, effect of worker weight ratio: $F_{1,204} = 0.003$, $P = 0.959$; Fig. 3c), or hornet counts at sites (GLMM, effect of hornet count: $F_{1,83} = 1.237$, $P = 0.269$). Instead, predation attempt frequency correlated with the amount of time that hornets spent foraging near colonies (GLMM, effect of hornet duration at colonies: $F_{1,204} = 12.410$, $P < 0.001$; Fig. 3b), independent of colony ID (colony random

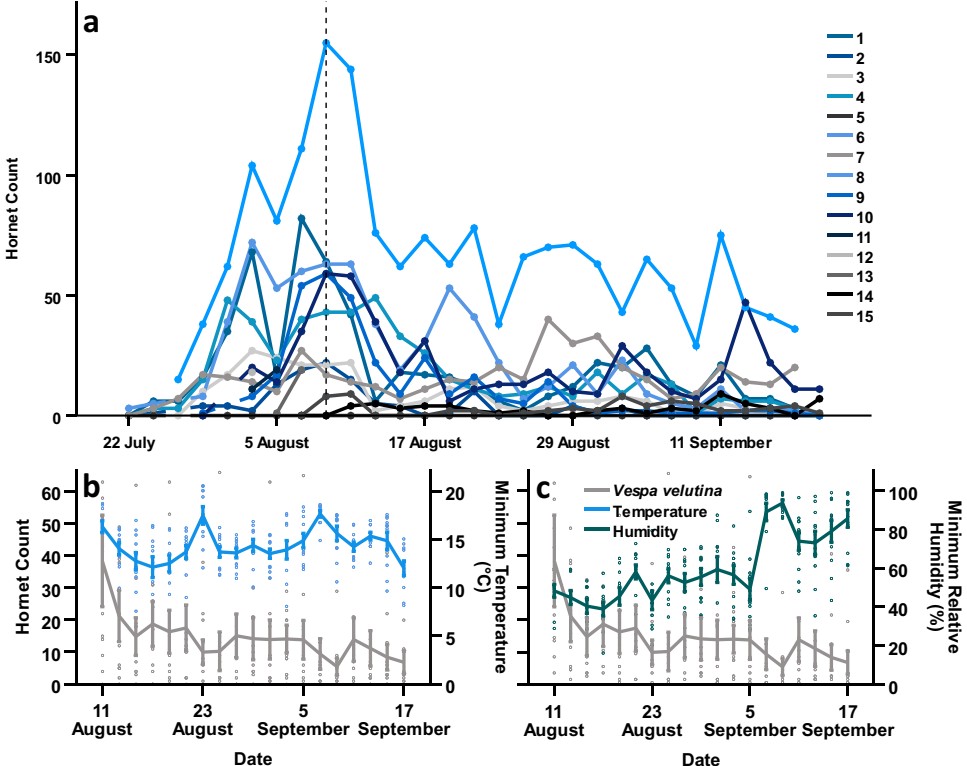

**Fig. 2 Hornet counts at sites. a** Hornet counts at sites over time, determined by the number of individuals caught in two VespaCatch traps every two days ($N = 5302$). Line colours indicate site IDs, the dashed vertical line denotes the timing of colony establishment, and points prior to this indicate those used for site characterisation. **b** Mean hornet counts and minimum daily temperature for all sites across the duration of the study ($N = 3798$). Each point represents a single measure of hornet count or minimum daily temperature at a site from the corresponding date. Lines indicate moving averages, and colours denote the measured variable (hornet count, grey; minimum daily temperature, blue). **c** Mean hornet counts and minimum daily relative humidity for all sites across the duration of the study ($N = 3798$). Each point represents a single measure of hornet count or minimum daily relative humidity at a site from the corresponding date. Lines indicate moving averages, and colours denote the measured variable (hornet count, grey; minimum daily relative humidity, green). Error bars represent 95% confidence intervals.

effect: $Z = 1.107$, $P = 0.268$), and was again inversely related to temperature (GLMM, effect of site temperature: $F_{1,204} = 15.121$, $P < 0.001$) and relative humidity (GLMM, effect of site relative humidity: $F_{1,204} = 19.724$, $P < 0.001$) at the time of sampling.

**Factors influencing bumblebee colony foraging activity**. Foraging activity was recorded using a calibrated Dragon Touch Vision 1 camera (Dragon Touch) positioned directly above the entrance of each colony (Fig. 1a). During sampling, this allowed for the filming of all bumblebees passing through the transparent entrance tube against a standardised high-contrast background (Fig. 1c), along with confirmation of foraging through the presence of pollen. Automated tracking of bumblebees was then achieved via a modified version of Camlytics (release v .2.2.5) AI-assisted tracking software, yielding individual trajectories and timestamps for all worker transits, along with their directionality (Fig. 1c, e, S3, and Supplementary Video 1).

The frequency of worker departures from colonies increased with the percentage of heterogeneous agriculture at sites (GLMM, effect of heterogeneous agriculture: $F_{1,8} = 6.182$, $P = 0.036$), peaking in mid-August (Fig. 4a), and was positively associated with the probability of subsequent hornet presence (GLMM, effect of hornet presence at colonies: $F_{1,130} = 15.097$, $P < 0.001$; Fig. 4b). Additionally, departure frequency decreased with higher relative humidity at the time of sampling (GLMM, effect of site relative humidity: $F_{1,60} = 133.816$, $P < 0.001$; Fig. 4c), while the percentage of discontinuous urban fabric had no discernible influence (GLMM, effect of discontinuous urban fabric:

$F_{1,8} = 1.720$, $P = 0.226$), and colony ID had a significant random effect (colony random effect: $Z = 2.023$, $P = 0.043$).

In a similar fashion, the frequency of worker returns to colonies decreased with relative humidity at the time of sampling (GLMM, effect of site relative humidity: $F_{1,51} = 104.780$, $P < 0.001$; Fig. 4c), peaked in mid-August (Fig. 4a), and was again positively associated with subsequent hornet presence (GLMM, effect of hornet presence at colonies: $F_{1,140} = 16.916$, $P < 0.001$; Fig. 4b). This suggests that hornets were attracted to colonies with higher foraging traffic (Fig. S4), but that their presence did not lead to substantial foraging inhibition, as is the case in honey bees[42]. Colony ID again had a significant random effect (colony random effect: $Z = 2.190$, $P = 0.029$).

**Factors influencing bumblebee colony growth and survival**. To assess the influence of factors upon colony growth and survival, we recorded colony weight change, survival status, and visible symptoms of disease across the sampling period. Then, upon study conclusion, we destructively sampled colonies to quantify the total adult population, mean weight of adult bumblebees, number of open and closed pupal cells, and the presence or absence of parasites, new queens, and males (Fig. S5). These measures were used to evaluate the site-level effects of hornet density, land cover, and climatic conditions upon colony growth and survival. Further, from a subset of 10 colonies, the prevalence of common pathogens was assessed, allowing us to characterise environmental pathogen exposure in visibly symptomatic and asymptomatic colonies (Supplementary Methods, Tables S2–S5).

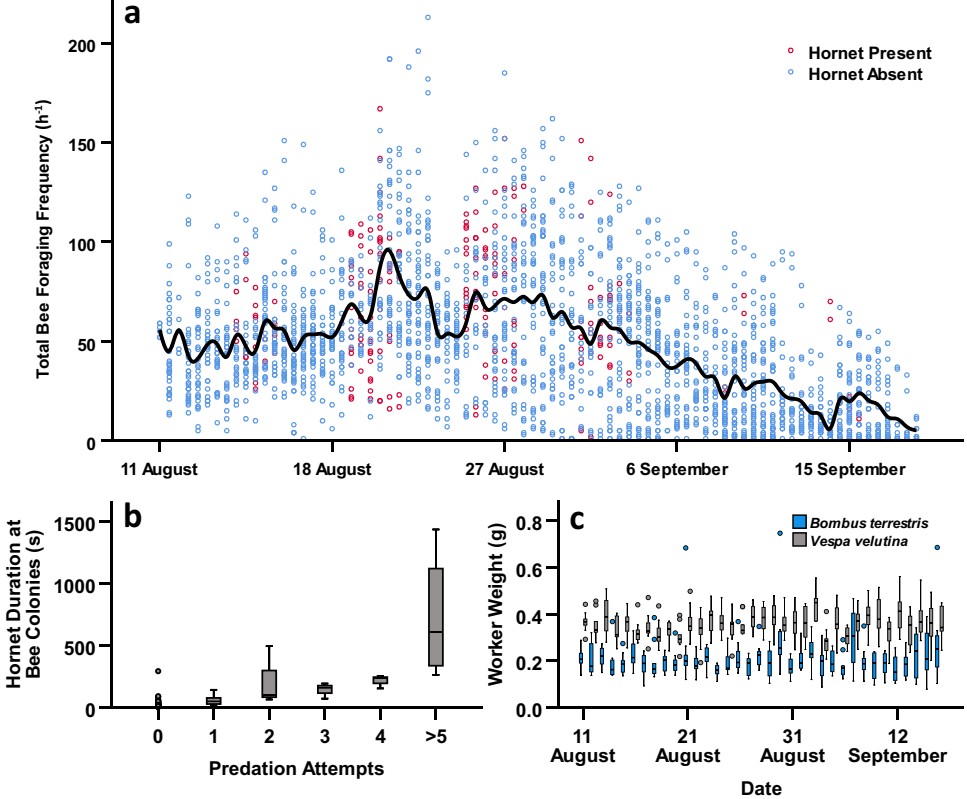

**Fig. 3 Hornet behaviour at bumblebee colonies. a** Total bee foraging frequencies for colonies over time ($N = 36$). Each point represents a single sample of bee foraging frequency per hour for a colony from the corresponding date. Points are coloured based on whether hornets were present at the colonies (absent, blue; present, red), and the solid line indicates a LOESS (locally weighted scatterplot smoothing) moving average for all colonies over time. **b** Time spent by hornets foraging at colonies ($N = 28$), grouped by the number of resultant predation attempts observed. Outliers (greater than 1.5 times the interquartile range from the median) are indicated with circles. **c** Weights of individual bee and hornet workers sampled across the course of the study ($N_{b.\ terrestris} = 350$, $N_{V.\ velutina} = 350$). Boxplots are coloured by species (*B. terrestris*, blue; *V. velutina*, grey) and grouped by sampling date. Outliers (greater than 1.5 times the interquartile range from the median) are again indicated with circles.

Hornet counts at sites had no significant influence on bumblebee colony survival (GLMM, effect of hornet count: $F_{1,676} = 0.474$, $P = 0.491$; Fig. 4), however a higher percentage of heterogeneous agriculture (GLMM, effect of heterogeneous agriculture: $F_{1,10} = 6.206$, $P = 0.032$), higher minimum relative humidity (GLMM, effect of minimum relative humidity: $F_{1,676} = 14.602$, $P < 0.001$), and higher minimum temperature (GLMM, effect of minimum temperature: $F_{1,676} = 10.225$, $P = 0.001$), increased the likelihood that colonies would survive. Colony ID also had a significant random effect when considering survival outcomes (colony random effect: $Z = 2.696$, $P = 0.007$).

In contrast, colony weight over time was negatively correlated with hornet counts (GLMM, effect of hornet count: $F_{1,653} = 83.175$, $P < 0.001$; Fig. 5), and higher maximum temperatures at sites (GLMM, effect of maximum temperature: $F_{1,640} = 9.548$, $P = 0.002$), while being positively influenced by higher minimum temperatures (GLMM, effect of minimum temperature: $F_{1,634} = 9.436$, $P = 0.002$). When considering these interactions, colony ID had a significant random effect (colony random effect: $Z = 3.952$, $P < 0.001$), indicating a degree of unexplained inter-colony variability, as would be expected.

Hornet counts had no significant effect on the production of new queens by colonies (GLMM, effect of hornet count: $F_{1,716} = 0.051$, $P = 0.821$), while a higher percentage of heterogeneous agriculture (GLMM, effect of heterogeneous agriculture: $F_{1,10} = 11.250$, $P = 0.007$) and water bodies at sites (GLMM, effect of water bodies: $F_{1,9} = 6.121$, $P = 0.035$) increased the

likelihood of new queens being present, independent of site ID (site random effect: $Z = 1.677$, $P = 0.094$).

Colony population size was positively correlated with colony weight across the study (GLMM, effect of colony weight: $F_{1,698} = 4.154$, $P = 0.042$). This relationship was independent of hornet counts (GLMM, effect of hornet count: $F_{1,698} = 0.428$, $P = 0.513$), and the percentage of heterogeneous agriculture at sites (GLMM, effect of heterogeneous agriculture: $F_{1,10} = 2.054$, $P = 0.182$), with site ID having a significant random effect (site random effect: $Z = 2.159$, $P = 0.031$). Similarly, the average weight of individual adult bumblebees in colonies was positively correlated with colony weight over time (GLMM, effect of colony weight: $F_{1,698} = 7.663$, $P = 0.006$), yet unaffected by hornet counts (GLMM, effect of hornet count: $F_{1,689} = 0.995$, $P = 0.319$), and the percentage of evergreen broadleaved and coniferous forests at sites (GLMM, effect of evergreen broadleaved and coniferous forests: $F_{1,10} = 3.362$, $P = 0.097$). When considering average adult bee weight, site ID again had a significant random effect (site random effect: $Z = 2.167$, $P = 0.030$).

## Discussion
Our results demonstrate that *V. velutina* workers frequently attempt to predate upon *B. terrestris* foragers at colony entrances, yet in the present study, we observed such attempts to be entirely unsuccessful. Hornets were preferentially attracted to colonies with a higher frequency of foraging traffic, and the duration of this attraction was significantly correlated with the probability of predation, suggesting that continuous prey flux is a requisite

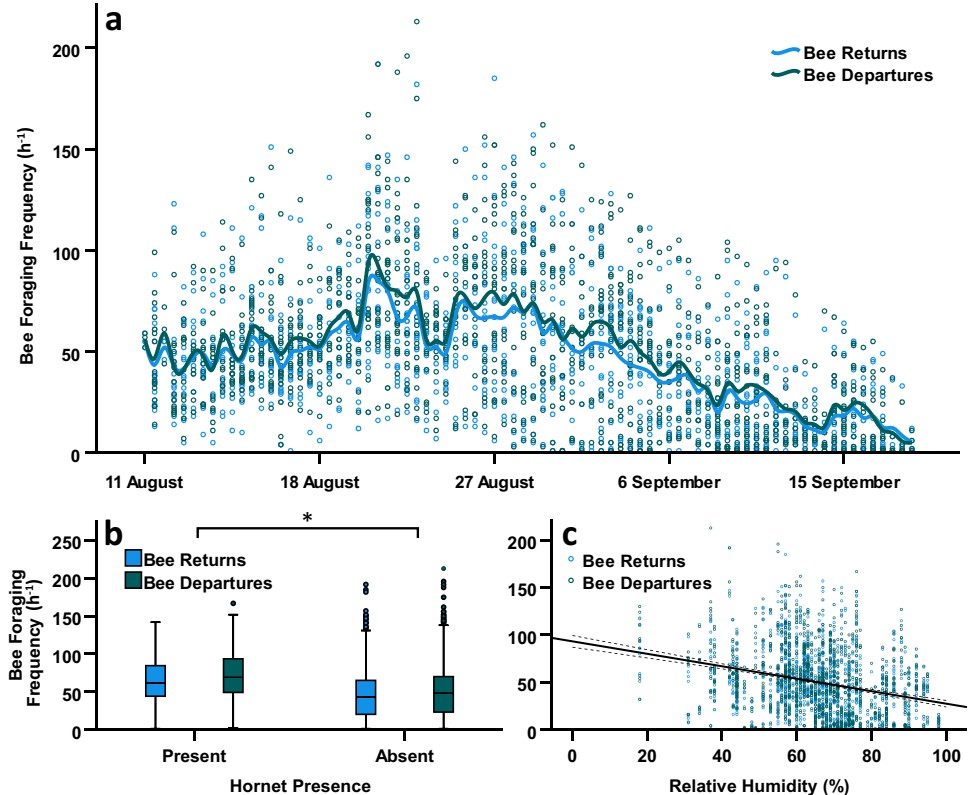

**Fig. 4 Bumblebee colony foraging activity. a** Frequency of bee departures and returns per hour for all colonies over time ($N = 36$). Each point represents a single measure of departure or return frequency per hour for a colony from the corresponding date. Lines indicate LOESS (locally weighted scatterplot smoothing) moving averages, and colours denote worker movement direction (returns, light blue; departures, dark blue). **b** Frequency differences in bee departures and returns per hour when hornets were present or absent ($N_{Present} = 28$, $N_{Absent} = 36$). Boxplots are coloured by worker movement direction (returns, light blue; departures, dark blue) and grouped by hornet presence or absence at colonies. Asterisk indicates significant differences between groups (entrances, $F_{1,130} = 15.097$, $P < 0.001$; exits, $F_{1,140} = 16.916$, $P < 0.001$). Outliers (greater than 1.5 times the interquartile range from the median) are denoted with circles. **c** Relationship between site relative humidity at the time of sampling and the frequency of bee returns and departures per hour across all colonies ($N = 36$). Each point represents a single measure of bee departure or return frequency per hour and the corresponding relative humidity at the time of sampling. Point colour denotes worker movement direction (returns, light blue; departures, dark blue), the solid line is a linear trendline fitted against the data, and the dashed lines represent the 95% confidence interval.

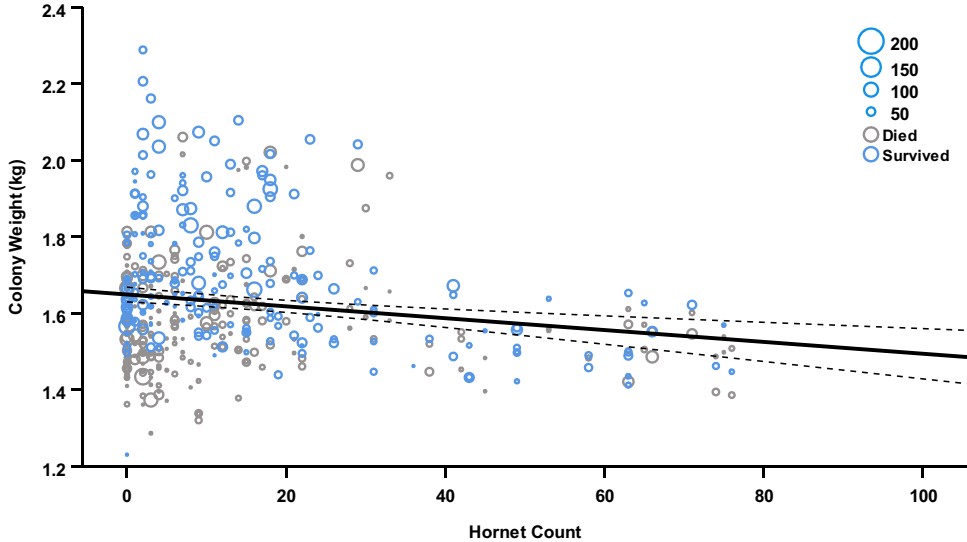

**Fig. 5 Bumblebee colony growth and survival.** Relationship between colony weight, total bee foraging frequency, and hornet counts at sites ($N = 36$). Each point represents a single measure of colony weight and the corresponding hornet count from two VespaCatch traps at the time of sampling ($N = 3798$). Point colour denotes the final survival outcome of the colony (blue, survived; grey died), while point size indicates the total bee foraging frequency at the time of colony weight sampling (larger, higher frequency). The solid line is a linear trendline fitted against the data, and the dashed lines represent the 95% confidence interval.

stimulus for the maintenance of predatory behaviour. Notably, bumblebee colony weights were negatively associated with hornet counts at sites, indicating possible collateral effects upon colony growth, ostensibly through foraging disruption via repeated predation attempts at or away from colonies, or competition for local resources. However, while colony growth was reduced, this did not appear to impact either survival or queen production, suggesting a degree of resilience. In concert, our results provide the first colony-level data quantifying the impacts of *V. velutina* on *B. terrestris*, and further elucidate the predatory behaviour of these invasive hornets, thus informing future mitigation efforts for wild and managed pollinators.

The repeated failure of hornets to successfully capture and subdue bumblebees is notable. Predation attempts generally followed a predictable sequence: a hornet would first pursue and grab a bee in flight (Fig. S6a–c); the bee would subsequently drop to the ground in response, causing the hornet to fall with it (Fig. S6d–f); and then, upon impact with the ground, the hornet would lose its grip, thus allowing the bee to escape (Fig. S6g–i and Supplementary Video 2). In some cases, a hornet was able to maintain its hold upon a bee after hitting the ground, however the bee would then assume a typical defensive posture, falling onto its back with legs and sting raised[38,43]. Without exception, this behaviour eventually forced the hornet to abandon the predation attempt, and return to foraging at the colony entrance. Interestingly, it is unlikely that such failure can be attributed solely to the comparative size of *B. terrestris* adults, as we observed predation attempts directed towards individuals spanning the range of worker polymorphism present in this species (Fig. 3c), all of which were unsuccessful.

Such interactions appear in stark contrast to predation targeting *A. mellifera*, with previous work finding that in ~30% of foraging visits to colonies, *V. velutina* were able to successfully capture, subdue, and kill workers[44]. Indeed, confirmed cases of the hornets predating upon *Bombus* species in their invaded range are rare to date, with limited visual reports[45,46], and a comprehensive dietary analysis finding only a single record of *B. lapidarius* among a sample of 2515 prey items[34]. Further, a recent study of hornets hunting at *Hedera hibernica* flowers observed agonistic interactions, but never a successful predation attempt towards wild *B. terrestris* queens, indicating that failed predation attempts are present in nature[47]. While the paucity of detections in other dietary analyses may be due to the limited period of sampling, and primer biases that suppress the amplification of bumblebee DNA[35], additional evidence for successful predation upon *Bombus* spp. is scarce, even within the native range of *V. velutina*[48,49]. Taken with our present findings, this suggests that *B. terrestris* is not a major prey species of *V. velutina*; however as noted in earlier work[36,48,49], this fails to deter the hornets from making repeated predation attempts.

While the impacts of predation per se appear minimal, our results do suggest that the presence of hornets—especially at very high densities—may impart a negative impact on bumblebee colony growth. The exact mechanisms underlying this remain unclear, thus it is impossible to completely rule out additional confounding factors. Notably, although our study accounted for the influence of land cover at a broad scale, it is feasible that the availability of specific resources within sites—such as floral nectar or other insect prey—gave rise to the observed trends. The local density of managed honey bee colonies may also have been a factor, as apiaries can suppress bumblebee colony growth;[50] however we were unable to collect this data within the confines of our study. While such effects are plausible, there are also several direct mechanisms by which hornet density may have impacted colony weight. Specifically, previous work has shown that bumblebees foraging at flowers alter their behaviour in response to

hornet attacks[36], concomitantly reducing the efficacy of resource collection. Furthermore, increased exploitation of floral nectar in areas with high hornet densities, combined with harassment of foraging bumblebees, would serve to reduce resource availability, while imposing energetic costs associated with repelling attacks. Indeed, data from honey bees indicates that the presence of hornets can impact colony health through reduced queen fecundity[33,51], and the increased expression of oxidative stress-related genes, even in bees that are not directly targeted[52]. Despite this, it is important to note that the observed reductions in bumblebee colony weight did not result in reduced colony survival or queen production over the course of the study. While this is promising, further work would be needed to rule out population-level risks over longer time scales, because such growth effects have the potential to accumulate across multiple generations.

Local land cover and climate appeared to be of variable importance between species. While hornet counts at sites were largely independent of land cover, bumblebee colony survival, queen production, and foraging activity were all positively correlated with the percentage of heterogeneous agriculture in the surrounding area. This is perhaps unsurprising, as *V. velutina* is a highly adaptable generalist forager[34], while *B. terrestris* is necessarily sensitive to the local availability of suitable pollen and nectar sources[53]. The negative correlation between hornet counts and water bodies at sites is less clear, and may be indicative of covariance with additional unmeasured parameters, however, as all water bodies were ≥0.1 km² in area, it may also suggest potential exclusion effects. Both species responded positively to higher daily minimum temperatures, while having a negative response to higher daily maximum temperatures, suggesting that these were limiting factors. Additionally, relative humidity at the time of sampling negatively influenced both hornet activity and bumblebee foraging frequency, as would be expected due to its tight association with precipitation.

Several colonies showed signs of disease during the study, namely dysentery, and results from the pathogen analyses were able to provide some further insight into the potential causative agent. Specifically, evidence of dysentery was correlated with the prevalence of the fungal pathogen *Vairimorpha bombi*—formerly *Nosema bombi*[54]—which is consistent with the observed symptomology (Fig. S7). It should be stressed that the limited sample size used here precluded any causative analyses, however as colonies were screened prior to establishment (Table S2), it is reasonable to deduce that *V. bombi* infections were acquired naturally, especially given its high prevalence in the wild[55].

The relationship observed between bee foraging frequency, hornet visits to colonies, and the frequency of predation attempts, provides some insight into the potential stimuli required to promote predatory behaviour in *V. velutina*. Specifically, hornets appear to require constant foraging traffic to elicit continued interest in a colony, and disengage once this frequency falls below a certain threshold (Fig. S4). The former point is salient, as it suggests that forager activity—rather than any other unique attribute of the experimental colonies—is a key driver of hornet attraction. This trend is consistent with the hornets' preference for other dense prey aggregations, including pollinators at flower patches;[36,48] flies around livestock and refuse;[34,45] and colonies of honey bees[42,44,56]. An important distinction when considering *B. terrestris*, however, is that colony foraging activity appears to be at the edge of the threshold required to maintain the hornets' interest, as evidenced by the frequent cessation of predatory behaviour at lower foraging frequencies (Fig. S4). Indeed, foraging activity showed little discernible response to hornet presence, supporting a causal influence of forager flux upon hornet attraction, rather than any defensive reaction by the colony.

Due to the need for weight standardisation, detailed colony assessments, and tracking of survival outcomes, we were constrained to the use of commercially-reared colonies in the present study. However, a notable feature of these colonies is their increased visibility when compared to natural *B. terrestris* nests. It is thus possible that wild colonies of this species are less attractive to foraging hornets, because they are often located underground[38], and may not always reach the forager densities observed in the present study. Despite this, as the flow of foraging traffic alone appeared to be the key determining factor in the hornets' attraction to colonies—regardless of the increased entrance visibility of our experimental setup—it can be reasonably surmised that natural colonies with sufficient foraging activity would be similarly targeted. Indeed, data from wild *B. terrestris* colonies indicates comparable worker populations to those recorded in the present study[57], with subsequent foraging activity being used by humans and predators to successfully locate nests[58]. Additionally, while our use of transparent entrance tubes increased the visibility of traversing bumblebees, it should be noted that hornet predation attempts generally occurred away from these, implying that their presence did not substantially alter hornet behaviour. As such, it can be postulated that our results are reasonably transferable to wild colonies, although further research would be needed to determine their importance in the context of natural populations; and for different bumblebee species.

The apparent preference of hornets for concentrations of prey may be pertinent to the protection of pollinating insects in invaded areas, as it provides impetus for limiting the cluster density of conservation measures that artificially aggregate solitary species[59]. Interestingly, the converse is indicated for managed honey bee colonies, because high densities are thought to 'dilute' the per capita predation burden[60]. The latter should be understood in the context of 'foraging paralysis' however, as the principle threat to honey bee colonies is their reaction to the intensity of predation, rather than direct mortality from predation itself[42,51]. Consequently, for species that do not undertake collective overwinter provisioning, including bumblebees, such effects are of limited applicability. This suggests that managed honey bee colonies should be considered as a special case, and for other less numerous aggregations of insects, strategies that diffuse local density are likely to be effective in reducing predation intensity.

In conclusion, we provide the first empirical data comprehensively quantifying the impacts of *V. velutina* upon *B. terrestris* colonies. Despite the finding that bumblebees were frequently targeted as prey, all such predation attempts that we observed were wholly ineffectual, highlighting the importance of detailed behavioural monitoring to ascertain outcomes. Further, the negative correlation between bumblebee colony weights and hornet counts at sites indicates that sub-lethal effects should be considered when evaluating colony-level risks, as this trend persists despite an apparent lack of predation-mediated mortality. Taken together, our findings suggest that *V. velutina* interacts quite distinctly with different bee genera, even within the eusocial Apidae alone. As such, there is strong precedent to consider threatened species in a bespoke fashion, if we are to effectively understand and mitigate invasion impacts at the ecosystem-level.

## Methods

**Colonies**. Colony preparation occurred prior to the initiation of the study during August 2021, near Tomiño in Pontevedra, Spain at 41.97903, −8.76852 (DD). To ensure standardisation, and facilitate controlled assessments, 36 coetaneous *Bombus terrestris terrestris* colonies were obtained from a local supplier in Almería, Spain (Biobest Group), each consisting of a queen and ~80 workers, and housed within waterproof polystyrene outer coatings (Biobest Group). Along with these, an additional three colonies were purchased for equipment testing and optimisation. All colonies were screened by the supplier for an extensive panel of pathogens and parasites using RT-qPCR, and the absence of infection was confirmed (Table S2). Upon delivery, colonies were weighed, assigned a random ID number, and provided with supplemental sucrose solution to assure adequate provisioning. Each was then allocated a site ID number, totalling three colonies per site, ensuring an even starting weight distribution across sites (mean weight <±50 g between sites, Fig. S8a). Following transportation to sites, supplemental sucrose was removed to stimulate foraging, and colony entrances were opened. Colonies were then allowed to acclimatise and familiarise themselves with their surroundings for 24–48 h prior to sampling.

**Experimental design**. Preceding colony setup, 15 prospective field sites were identified throughout the province of Pontevedra, Spain. These sites were selected to include a range of local *Vespa velutina nigrithorax* densities and land cover types, allowing us to evaluate colony outcomes in a representative environment. To characterise relative hornet abundance and activity, we deployed two VespaCatch traps with VespaCatch attractant (Véto-pharma) at each site[61], at a height of 1.5 m from the ground, and out of direct sunlight. Foam inserts were placed into all traps to absorb excess liquid attractant and prevent captured insects from drowning, thus limiting mortality and unwanted effects upon both native species and hornets[62]. Captured hornets were counted and released every second day for the two weeks preceding study initiation in late July and early August 2021. The resultant measure consisted of the total number of hornets caught in both traps at a site over two days—henceforth termed 'hornet count'. This provided a profile of hornet activity over time, allowing us to select a final group of 12 field sites that best-encompassed the range of local hornet densities. Across the course of the study, we continued monitoring hornet counts at sites to ensure that our data remained temporally relevant, as activity can fluctuate substantially over time.

Each site was provisioned with a water source and sucrose feeder, allowing the colonies to forage *ad libitum*, while ensuring that water and carbohydrate availability were not limiting factors —although colonies were rarely observed utilising these. Colonies were situated in the shade amongst vegetation to avoid overheating, and in cases where sufficient natural shade was not present, were provided with shade in the form of pitched rooves of white Correx sheeting (Corplex UK). Experimental sampling was initiated on the 9th of August 2021, and continued for a period of 40 days, concluding on the 18th of September.

**Sampling**. Colony sampling occurred every two days for the duration of the study, producing 20 time points in total. Specifically, colonies were weighed using a PS 2100.R2.M.H precision (linearity ±0.02 g) balance (RADWAG), checked for visible symptoms of disease, and their survival status was noted. To quantify individual behaviour, two Dragon Touch Vision 1 cameras (Dragon Touch) were trained on each colony. The first of these was an 'entrance camera' situated 110 mm above a customised transparent acrylic entrance tube, yielding a frame area of ~0.011 m², and thus recording all bumblebees entering and exiting the colony during sampling (Fig. 1a). The second was an 'external camera' positioned 1 m from the colony, with its field of view encompassing the exterior and proximate surroundings within ~1.5 m, enabling observation of all hornets in the immediate vicinity, and thus allowing predation behaviour to be

recorded (Fig. 1b). Video recordings occurred between 09:00 and 17:00 h to best align with colony foraging activity[63], and averaged 3–4 h in length, dependent on battery life (Fig. S8b). Spent cameras were collected and replaced at each time point, allowing data retrieval and initiation of recording to occur simultaneously. This rotation employed a total of 72 cameras, enabling complete coverage for all 36 colonies.

In addition to colony assessments, continuous site-level temperature (°C) and relative humidity (%) measurements were made using Tinytag Plus 2 - TGP-4500 data loggers (Gemini Data Loggers Ltd). These recorded values at 15 min intervals, enabling the calculation of daily maxima and minima, along with ambient conditions at the time of video recordings. To characterise land cover, satellite data was extracted from a 1.5 km radius around each site, utilising the 2018 CORINE[64] land cover dataset in ArcGIS Pro (release v. 3.0) (Fig. S2). This provided land cover data encompassing the maximum foraging range of colonies[65] to a resolution of 100m$^2$, which was then validated against recent satellite imagery to produce modified class definitions that more accurately represented the local vegetation (Fig. S3 and Table S1).

Further, from a single site per day, 10 bee and 10 hornet foragers were caught and weighed using HSW07 precision (linearity ±0.001 g) scales (Hoosiwee), thus comparing the weight ratio between species over time, to determine whether this influenced predation dynamics. Ratios were calculated using the following formula:

Worker weight ratio = mean hornet weight(g)/mean bee weight(g)

$$(1)$$

Upon conclusion of the study, colonies were destructively sampled with ethyl acetate to quantify the total adult population, mean weight of adult bumblebees (g), number of open and closed pupal cells, colony weight (g) and dimensions (mm), and the presence or absence of gynes, males, nectar, and pollen. Males were identified via extrusion of the endophallus, while gynes were differentiated by mapping their position relative to known queens in the overall weight (g) frequency distribution (Fig. S5). Additionally, the presence or absence of bumblebee wax moth larvae (*Aphomia sociella*) was recorded, along with those of the black soldier fly (*Hermetia illucens*).

To investigate environmental pathogen exposure, a subset of 10 colonies was screened for common fungal, trypanosomal, and neogregarine pathogens. This subset consisted of five colonies showing visible symptoms of disease—specifically dysentery—and five showing no visible symptoms. Subsequent analyses were descriptive rather than inferential, and aimed to characterise the pathogen profiles of symptomatic and asymptomatic colonies. A panel of seven targets was chosen, consisting of *Vairimorpha ceranae*, *Vairimorpha apis*, *Vairimorpha bombi*, *Crithidia bombi*, *Crithidia mellificae*, *Lotmaria passim*, and *Apicystis bombi* (Table S3). Analyses were conducted using a combination of multiplex and monoplex PCR, and DNA extraction protocols followed previously established methods[60]. For full details of the PCR and DNA extraction methodologies, primer sequences, and thermal protocols, see (Supplementary Methods, Tables S4, S5).

**Video analyses**. Video analyses utilised two complementary processing techniques. Videos from the entrance cameras were analysed via automated tracking software, while those from the external cameras were processed semi-automatically using event-logging software (Fig. 1c, d). Analyses were dispersed evenly across the 40 day sampling period, with the frequency of sampling being constrained by camera rotation, as every third day all 72 cameras were deployed simultaneously. This yielded 13 time

points of data comprising 135,983 individual trajectories from the entrance cameras, and 6 time points of data comprising 648 h of behavioural data from the external cameras. Prior to processing, all videos were normalised to a standard length of 3 h, and timestamps combined with an audio cue were used to synchronise timings between cameras.

*Automated tracking*. To assess whether hornet behaviour had an impact on foraging activity, we used a modified version of Camlytics (release v .2.2.5) AI-assisted tracking software to track individual bumblebees entering and leaving colonies through the observation entrances (Fig. 1c, S3, and Supplementary Video 1). The custom design of these entrances ensured that bumblebees remained on a fixed plane of movement, while a white plastic insert below the entrance simplified the background to optimise detection (Fig. 1e). Further, in comparison to standard commercial nest boxes, the extended entrance length served to better approximate the tunnel entrances present in nature[57]. Colonies were marked with calibration and range-finding indicators above the entrances, ensuring that all cameras were mounted at a consistent position and distance, thus allowing serial processing of videos (Fig. 1a). When a bee traversed an entrance, it was individually identified as an object and its trajectory was tracked, enabling movement direction to be determined. This was then detected by a digital 'entrance counter' region, allowing the exact timing and number of entrances and exits to be recorded (Fig. 1c, S3, and Supplementary Video 1). From the resultant data, three key metrics were extracted, frequency of bee departures, defined as the number of bee exits per hour; frequency of bee returns, defined as the number of bee entrances per hour; and total bee foraging frequency, this being calculated as the sum of departures and returns per hour, and confirmed via the presence of pollen carried by returning bumblebees. To validate the system, a subset of videos were then analysed manually, ensuring concordance between results.

*Semi-automated observation*. To assess whether hornets attempted to predate upon bumblebees at colony entrances, videos from the external cameras were analysed using BORIS (release v. 8.5)[66], allowing observers to record a suite of prespecified events and behaviours (Fig. 1f). Specifically, these included hornet presence at colonies, defined as hornets foraging in proximity to a colony; hornet duration at colonies, defined as the total time spent (s) by hornets in proximity to a colony; predation attempts, defined as hornets pursuing and subsequently making physical contact with bumblebees; predation successes, defined as hornets subsequently catching bumblebees; and predation failures, defined as hornets failing to catch a bee after making an attempt. Observers recorded these events manually and the software subsequently extracted behaviour type, timing, frequency, and duration data (Fig. 1d). Prior to and during these analyses, interobserver consistency was validated using the BORIS inter-rater reliability function.

**Range of assessment factors**. Measured variables were broadly grouped into those of hornet abundance and behaviour, colony growth, survival, and foraging activity, site climate, and land cover. Within these groups, we examined a combination of colony-level, individual-level, and site-level parameters.

Hornet abundance and behaviour were assessed using hornet counts at sites over time, hornet presence at colonies, hornet duration at colonies, predation attempts, predation successes, predation failures, and worker weight ratio.

Bumblebee colony growth and survival were evaluated using colony weight, colony survival status, total adult population, mean weight of adult bumblebees, number of open and closed

pupal cells, and the presence or absence of gynes, males, wax moths, nectar, and pollen.

The foraging activity of colonies was quantified using the frequency of bee departures, frequency of bee returns, and total bee foraging frequency.

Local climatic conditions were measured using daily minimum and maximum temperature, daily minimum and maximum relative humidity, site temperature at the time of sampling, and site relative humidity at the time of sampling.

Finally, site land cover was characterised by the percentage cover of nine land classes, including deciduous broadleaved forest, discontinuous urban fabric, evergreen broadleaved and coniferous forest, heterogeneous agriculture, natural grass-lands, moors and heathland, transitional woodland shrub, vineyards, and water bodies (for full land class definitions, see Table S1).

**Statistical analyses**. To determine the subset of variables to be included in final analyses, we employed random forest variable importance rankings using the R package 'randomForest'[67]. From the initial set of measured variables, importance was ranked by mean contribution to the Gini index for categorical responses, and contribution to node purity for numerical responses (Fig. S9). All models were optimised by plotting the out-of-bag and mean square error rates to confirm stabilisation, using the R packages 'data.table'[68], 'cowplot'[69], and 'ggplot2'[70].

To assess factors influencing hornet counts at sites, their behaviour at colonies, and resultant effects upon colony growth, survival, and foraging activity, we utilised generalised linear mixed models (GLMMs) with repeated measures. Separate models were generated for each response variable, to maximise sample sizes while minimising model complexity. Hornet counts and colony assessments were repeated across all 20 time points, while automated tracking videos and semi-automated observation videos were recorded for 13 and 6 evenly dispersed time points, respectively. Model selection was based on AIC, beginning with the full model and interactions, and degrees of freedom were calculated using the Welch–Satterthwaite approximation. Model fit was validated via evaluation of the binned standardised or Pearson residuals.

Inter-rater reliability in BORIS analyses was assessed using Cohen's kappa coefficient, requiring a minimum concordance ($\kappa$) of 0.70 between observers at a resolution of 1 s. Across tests, we confirmed requisite sample sizes to provide a minimum power ($1-\beta$) of 0.80, at an alpha ($\alpha$) of 0.05, using standard deviation ($\sigma$) and mean difference ($\delta$) values from the data. Statistical analyses were performed in SPSS (release v. 28.0.1.1) and R (release v. 4.2.1)[71].

*Factors influencing hornet counts at sites*. The GLMM assessing the determinants of hornet counts at sites used hornet count as a Poisson response variable with a log link, minimum temperature, minimum relative humidity, date, percentage of discontinuous urban fabric, and percentage of water bodies as fixed factor predictors, and site ID as a random factor.

*Factors influencing hornet activity at bumblebee colonies*. The GLMM assessing factors affecting hornet presence at colonies used hornet presence at colonies as a binomial response variable with a probit link, colony weight, total bee foraging frequency, site temperature at the time of sampling, and site relative humidity at the time of sampling as fixed factor predictors, and colony ID nested within site ID as random factors. The GLMM assessing factors influencing the frequency of hornet predation attempts used predation attempts as a Poisson response variable with a log link, hornet count, hornet duration at colonies, worker weight

ratio, site temperature at the time of sampling, and site relative humidity at the time of sampling as fixed factor predictors, and colony ID nested within site ID as random factors.

*Factors influencing bumblebee colony growth and survival*. The GLMM assessing the effect of hornet counts on colony survival used colony survival status as a binomial response variable with a probit link, hornet count, minimum temperature, minimum relative humidity, and percentage of heterogeneous agriculture as fixed factor predictors, and colony ID nested with site ID as random factors. The GLMM assessing the effect of hornet counts on colony weight used colony weight as a gamma response variable with a log link, hornet count, maximum temperature, and minimum temperature as fixed factor predictors, and colony ID nested within site ID as random factors with a first-order autoregressive covariance structure to account for response autocorrelation.

The GLMM assessing the effect of hornet counts on colony queen production used the presence of new queens as a binomial response variable with a logit link, hornet count, percentage of heterogeneous agriculture, and percentage of water bodies as fixed factor predictors, and site ID as a random factor. The GLMM assessing the effect of hornet counts on colony population size used the total number of adults in colonies as a negative binomial response variable with a log link, hornet count, colony weight, and percentage of heterogeneous agriculture as fixed factor predictors, and site ID as a random factor. The GLMM assessing the effect of hornet counts on adult bee weight used adult weight as a normal response variable with an identity link, hornet count, colony weight, and percentage of evergreen broadleaved and coniferous forest as fixed factor predictors, and site ID as a random factor.

*Factors influencing bumblebee colony foraging activity*. The GLMM assessing the effect of hornet presence at colonies on the frequency of worker departures used the frequency of bee departures as a normal response variable with an identity link, hornet presence at colonies, site relative humidity at the time of sampling, percentage of heterogeneous agriculture, and percentage of discontinuous urban fabric as fixed factor predictors, and colony ID nested within site ID as random factors. The GLMM assessing the effect of hornet presence at colonies on the frequency of worker returns used the frequency of bee returns as a gamma response variable with a log link, hornet presence at colonies and site relative humidity at the time of sampling as fixed factor predictors, and colony ID nested within site ID as random factors.

**Statistics and reproducibility**. Analyses employed a sample population of 36 *B. terrestris* colonies, assigned to field sites in triplicate, across a total of 12 sites. Foraging analyses utilised a sample of 135,983 individual trajectories, and weight ratio assessments were based on data from 700 individuals. Pathogen analyses were conducted on a subset of 10 colonies, each yielding a sample of 20 individuals, and final colony assessments were performed on all 36 colonies, incorporating 940 individuals. The source data underlying all figures and analyses are available within the supplementary data. Full details of statistical tests, subset sample sizes, and variable selection procedures are provided in the results and statistical analyses sections.

**Reporting summary**. Further information on research design is available in the Nature Portfolio Reporting Summary linked to this article.

## Data availability

The authors declare that all supporting data is available within the supplementary information. For source data underlying the figures and analyses, see (Supplementary Data).

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

## Acknowledgements

T.A.O.-W, R.J.C, J.P, P.J.K, and J.L.O were funded by the Biotechnology and Biological Sciences Research Council (BBSRC) (Project No. BB/S015523/1). P.J.K, S.V.R.-N, X.M, C.B, and D.D.-M were funded by the European Regional Development Fund (ERDF) Interreg Atlantic Area Programme (Project No. EAPA_800/2018). The authors wish to thank Jaime Piñeiro, Cristina Martínez, Francis Martínez, Ana Belén Casaleiro, Rosendo González, Indalecio Bastos, Francisco Ramil, Andrés Domínguez, Iria Villar, and Dieter Boisits for providing field sites and technical assistance; and the University of Vigo for contributing laboratory facilities, equipment, and logistical support.

## Author contributions

T.A.O.-W, J.P, P.J.K, and J.L.O conceived the study; T.A.O.-W wrote the manuscript; S.V.R.-N and S.M located field sites; T.A.O.-W, R.J.C, E.K.J.G, D.S.R, and P.J.K collected the data; X.M, C.B, and D.D.-M conducted the pathogen analyses; and T.A.O.-W carried out the statistical analyses. All authors edited the manuscript, and gave final approval for publication.

## Competing interests

The authors declare no competing interests.
