## [Peer Review File · Communications Biology]

Reviewers' comments:

Reviewer #1 (Remarks to the Author):

The manuscript by O'Shea-Wheller and coauthors investigates the potential impact of the invasive yellow-legged hornet, *Vespa velutina nigrithorax*, on commercial colonies of the bumblebee *Bombus terrestris*.

I have found the topic of the manuscript very interesting and timely, since we mostly lack studies on the impact of this alien predator on wild insect pollinators and entomofauna in general.

The authors carried out a lot of work in the study, collecting a large array of different data on a pretty high number of experimental colonies in the field (N=36). The experimental procedure and the methods are clearly explained and easy to follow. However, I have some major concerns about the premises and the results of the study, which may be somehow linked, and the authors should carefully address them to clarify the rationale of their experiment and the results' validity and generalization under natural conditions.

First of all, the authors report that the effects of *V. velutina* on "natural pollinator populations remain poorly understood". Nonetheless, in the paper, the effects of *V. velutina* is assessed on artificial colonies of the commercial bumblebee, *B. terrestris*, and not on naturally occurring colonies of bumblebees in general. Thus, the title of the manuscript might be adjusted to avoid a generalization to entirely natural conditions and other native species of bumblebees.

In fact, the use of artificial bumblebee colonies placed in the field by the authors to assess the hornet impact may be partially responsible for the apparently awkward results regarding the total inability of *V. velutina* foragers to successfully prey on bumblebees.

The white colony boxes could be a signal resembling honeybee hives in apiaries, a likely research image for foraging hornets, and the continuous traffic of bumblebee foragers from the visibly evident entrance might further attract hornets looking for prey. I am not sure that the entrance of natural colonies of bumblebees, usually well concealed in the undergrowth, would in fact be as attractive towards foraging hornets with respect to what documented for the artificial nest boxes in the present study.

I think that an assessment of the hornet foraging behaviour towards natural occurring bumblebee colonies is a crucial missing piece of information to strengthen the results of the study and to demonstrate that the observed foraging behaviour of *V. velutina* with respect to artificial nest boxes actually reflects the predatory impact of hornets on wild bumblebee colonies.

If the predator is not able to detect a well-hidden prey nest, it is unlikely that it will be attracted to it and will not produce a strong impact on the nest. Similarly, if predation on bumblebee colonies does not occur in the field, it is unlikely that the predator will experience any selective pressure to evolve an effective strategy to kill its prey. Thus, an assessment of the natural occurrence of predatory attempts of *V. velutina* on wild bumblebee colonies would be necessary to demonstrate that what observed in the study faithfully reproduce what happens in nature.

These observations are also linked to the result that all the 125 predatory attempts of hornet foragers recorded in the study were invariably unsuccessful. This is quite puzzling. Why should a predator keep trying to catch a prey, which defences cannot overcome?

Foraging hornets were captured, counted, and released at each nest site, but it is not specified if captured hornets were marked before release, therefore it is not clear if the same hornets kept coming to the artificial nest boxes to try in vain to catch a prey that they were apparently unable to kill. It seems then that besieging bumblebee nests (in the form of artificial nest boxes) would represent only a cost for hornets and once more I wonder if the visual resemblance with a honeybee hive, an ideal foraging situation for *V. velutina*, may not play a role in hornet attraction to these artificial nests. The authors themselves report that "hornets were preferentially attracted to colonies with a higher frequency of foraging traffic, and the duration of this attraction was significantly correlated with the probability of predation, suggesting that continuous prey flux is a requisite stimulus for the maintenance of predatory behaviour" (Discussion, lines 447-450). It is therefore possible that the foraging traffic, made conspicuous by the entrance of the white nest boxes, might cause an experimental bias attracting hornets to an unrewarding artificial resource at an unnaturally realistic

rate compared to wild bumblebee nests.

Overall, I acknowledge the considerable effort and experimental work of the authors, but I feel that further information and clarifications concerning the actual occurrence and impact of interactions between *V. velutina* foragers and wild bumblebee colonies in the field are needed to corroborate and give the right credit to the results of the study.

Reviewer #2 (Remarks to the Author):

Dear colleagues,

I reviewed the manuscript "Quantifying the Impact of an Invasive Hornet on Bumblebee Colonies" (O'Shea-Wheller et al.). This manuscript presents a very interesting work about the predation of *V. velutina* on Bumblebee colonies. Protocols are well done and the obtained results are very interesting to colleagues interested by the impacts of this invasive species. The manuscript is well written and easy to read.

I have some minor comments that the authors need to address to improve (a little) their manuscript.

Materials and methods:

- Line 102: Could the authors give to the readers more information about pathogens and parasites analyzed and about the methods used please?

Discussion:

- Line 455: I disagree with the authors, as Rome et al. (2011) showed that very few Bumblebees were predated by the hornet. They need to modify the sentence and to give the reference please.

- Lines 479-490: The discussion is interesting and the work needs further experiments to clarify the effects of the *velutina* presence in front of bee colonies. However, the stress of colonies by *velutina* could greatly impact bumblebees. More and more works demonstrate this effect on honeybees. I think it could be more or less the same in bumblebee colonies. Authors need to discuss this point. For example, there is the work of Leza et al (2019).

- Line 521: A question, is it possible to estimate the threshold?

Very nice work! Thanks.

Sincerely

Eric

Reviewer #3 (Remarks to the Author):

This manuscript examines the important question of how an invasive hornet, *V. velutina nigrithorax*, impacts the fitness of a native bee species, the bumble bee, *Bombus terrestris*. The authors have taken great pains to replicate their experiment at multiple sites and to measure multiple variables, including bee health, bee fitness, bee and hornet behavior, and weather variables. The main findings are that 1) hornets tried to attack bees, but did not succeed; 2) in areas with more hornets, colonies weighed less, indicating lower fitness; and 3) hornet presence correlates with increased bee departures and returns. I think that the paper is well written (although the Discussion could be shortened, see below) and the data are correctly analyzed.

I think that this data is certainly worthy of publication, but I think that the authors should more fully explore the possibility that some of these correlations are due to other factors, such as the landscape which could drive both hornet abundance and bee colony variables. For example, please discuss hornet abundance may not necessarily drive down colony weights. Perhaps there is another factor such as other prey items? *Bombus terrestris* does not comprise much of the hornet diet (see cited

genetic analyses). If so, hornet abundance may be linked to their actual prey, including honey bee colonies. Adding this honey bee abundance to the models might be revealing.

I am not sure that it is correct to call the departures and returns at the nest entrance "foraging". It is interesting that bee departures and returns were higher when there were hornets, so perhaps what is counted is not actually foraging but alarm behavior? This possibility should be mentioned.

With respect to the colony setup (Fig. 1), is it possible that the tube used to monitor bee behavior also helped to prevent hornets from capturing bees? Such devices have been used to lower the success rate of *V. pensylvanica* entering and attacking honey bee nests. Attacks were also measured outside of the colony, but the potential effect of the tube is perhaps worth a mention.

Finally, it would be good for the authors to clarify the expected link between pathogens and hornet presence. Is the idea that hornets will stress bee foragers and thereby increase their pathogen load? Since you have a limited sample size that is too small for a causal analysis, I suggest that this aspect of the paper be placed in the SI. This will also help reduce the length of the Discussion.

Other comments

L102 please list the pathogens and parasites that your screen could identify.

L110 a map of the sites would be useful in the SI

L113 please provide citation showing that VespaCatch can capture *V. velutina nigrithorax*.

L452 Here you note that colonies were lighter in areas with more hornets and attribute this to foraging disruption. This supports the idea that bee departures and returns are not foraging but actually alarm.

L546 Please expand upon how this negative correlation indicates that sublethal effects should be considered.

Response to Referees

Editing Key:

Corresponding lines in revised manuscript-**yellow**, Replies-**blue**

Editorial Board Member, Dr Luke R. Grinham

Comments to Author:

Your manuscript entitled "Quantifying the Impact of an Invasive Hornet on Bumblebee Colonies" has now been seen by 3 referees. You will see from their comments below that while they find your work of considerable interest, some important points are raised. We are interested in the possibility of publishing your study in *Communications Biology*, but would like to consider your response to these concerns in the form of a revised manuscript before we make a final decision on publication.

Reviewer(s)' Comments to Author:

Referee: 1

The manuscript by O'Shea-Wheller and co-authors investigates the potential impact of the invasive yellow-legged hornet, *Vespa velutina nigrithorax*, on commercial colonies of the bumblebee *Bombus terrestris*.

I have found the topic of the manuscript very interesting and timely, since we mostly lack studies on the impact of this alien predator on wild insect pollinators and entomofauna in general.

The authors carried out a lot of work in the study, collecting a large array of different data on a pretty high number of experimental colonies in the field (N=36). The experimental procedure and the methods are clearly explained and easy to follow. However, I have some major concerns about the premises and the results of the study, which may be somehow linked, and the authors should carefully address them to clarify the rationale of their experiment and the results' validity and generalization under natural conditions.

First of all, the authors report that the effects of *V. velutina* on "natural pollinator populations remain poorly understood". Nonetheless, in the paper, the effects of *V. velutina* is assessed on artificial colonies of the commercial bumblebee, *B. terrestris*, and not on naturally occurring colonies of bumblebees in general. Thus, the title of the manuscript might be adjusted to avoid a generalization to entirely natural conditions and other native species of bumblebees.

Thank you for these constructive comments, we have amended the title to reflect the specific species of *Bombus* assessed in the study, and clarified the use of standardised commercially-reared colonies in the abstract and throughout. See lines: **1-2, 20, 86-87, 348-349, 357-358.**

In fact, the use of artificial bumblebee colonies placed in the field by the authors to assess the hornet impact may be partially responsible for the apparently awkward results regarding the total inability of *V. velutina* foragers to successfully prey on bumblebees.

Unfortunately, it would not have been feasible to find, and experimentally investigate, an appropriate number of size and age-matched wild bumblebee colonies in the area. This is due to the difficulty of locating nests, ensuring standardised starting weights, and conducting the requisite health screening. We also considered that the ultimate need to excavate and destroy them for evaluation would be unnecessarily harmful to the wild population, and thus antithetical to the aims

of our research. We thus opted to use standardised colonies in our experiment, but in a natural setting, and with natural levels of *V. velutina*.

While it is reasonable to question if our experimental setup would have affected the behaviour of hornets, comparable results have been observed in recent work assessing predation on wild bumblebees visiting ivy flowers (Rojas-Nossa et al. 2023). This suggests that the failure of hornets to successfully predate upon bumblebees in our study cannot solely be attributed to the use of artificial colonies. Indeed, the relative paucity of bumblebee prey samples in previous dietary analyses serves to further support this (Rome et al. 2021; Perrard et al. 2009). We have now added this citation and provided further details in the discussion. See lines: 297-298.

The white colony boxes could be a signal resembling honeybee hives in apiaries, a likely research image for foraging hornets, and the continuous traffic of bumblebee foragers from the visibly evident entrance might further attract hornets looking for prey. I am not sure that the entrance of natural colonies of bumblebees, usually well concealed in the undergrowth, would in fact be as attractive towards foraging hornets with respect to what documented for the artificial nest boxes in the present study.

It is unlikely that the white colony boxes are seen by hornets as resembling honey bee hives at apiaries, as in the study region of Pontevedra, hives are usually painted green, brown, or grey, rather than white, and are a different size and shape, giving them a distinct appearance. Further, as evidenced by the lack of interest in colonies with little foraging traffic, it is not the colony boxes themselves that appear to attract the hornets, but rather the flux of insect movement associated with foraging, as is the case in other natural prey aggregations (the hornets are known to target clusters of flies at rubbish tips; and pollinators at busy forage patches). It is also pertinent to note that the entrances of the artificial colonies used in our study were close to ground-level, and generally concealed in the shade of surrounding undergrowth. As both these entrances, and those of wild colonies, require bees to enter and exit in single file, it is also unlikely that the specific design of the entrances themselves seriously altered foraging traffic. We have now clarified the reasoning behind the entrance design, and provided additional discussion detailing the comparative attractiveness of artificial and wild colonies. See lines: 348-349, 362-365, 494-495.

I think that an assessment of the hornet foraging behaviour towards natural occurring bumblebee colonies is a crucial missing piece of information to strengthen the results of the study and to demonstrate that the observed foraging behaviour of *V. velutina* with respect to artificial nest boxes actually reflects the predatory impact of hornets on wild bumblebee colonies.

If the predator is not able to detect a well-hidden prey nest, it is unlikely that it will be attracted to it and will not produce a strong impact on the nest. Similarly, if predation on bumblebee colonies does not occur in the field, it is unlikely that the predator will experience any selective pressure to evolve an effective strategy to kill its prey. Thus, an assessment of the natural occurrence of predatory attempts of *V. velutina* on wild bumblebee colonies would be necessary to demonstrate that what observed in the study faithfully reproduce what happens in nature.

Unfortunately, such information is currently unavailable. It would be useful, however the difficulty of finding wild colonies in sufficient numbers precludes experimental tractability. While our study is the first to assess activity around colonies—albeit standardised ones—in a natural setting, there is data relating to the predation behaviour of *V. velutina* towards wild *Bombus terrestris* queens (Rojas-Nossa et al. 2023), and several other *Bombus* species (Williams 1988). Notably, while successful predation events are scarce, attempted predation is comparatively common, as would be expected

from the hornets' generalist hunting strategy. We have now included this information in the text, as these studies corroborate our own finding that hornet predation attempts directed at bumblebees are rarely successful. See lines: 293-298.

Vespa velutina is an adaptable and opportunistic predator, and previous studies indicate that it will be attracted to any congregation of potential prey. As wild bumblebee colonies can be located by human observers and predators through the identification of foraging traffic (Osborne et al. 2008), there is strong impetus to suggest that foraging hornets will also respond to this. We have now expanded upon this in the discussion. See lines: 348-351, 364-367.

These observations are also linked to the result that all the 125 predatory attempts of hornet foragers recorded in the study were invariably unsuccessful. This is quite puzzling. Why should a predator keep trying to catch a prey, which defences cannot overcome?

This is an interesting point when considering the behaviour of *V. velutina*. While hornets were initially successful in catching bumblebees in flight, it was their subsequent defensive response that led to eventual predation failure. Specifically, bees dropped from the air upon contact with a hornet, which generally caused the hornet to lose its purchase on impact with the ground. We have detailed this in the discussion, and provided a supplementary figure and associated video demonstrating the process in detail. See lines: 279-287, see figure: S6, see supplementary video: 2. There is also a substantial body of evidence to indicate that the hornets frequently attempt and fail to capture certain insects, with varying levels of success depending on the species (Tan et al. 2007; Rojas-Nossa and Calviño-Cancela 2020). It appears, however, that the payoff of rare but eventual success is sufficient to maintain the interest of hornets, and hence elicit repeated predation attempts. Indeed, the responses of other bumblebee species may differ, making them more or less viable predation targets. We have now outlined this in the discussion, and incorporated further references for context. See lines: 295-298, 300-301, 372.

Foraging hornets were captured, counted, and released at each nest site, but it is not specified if captured hornets were marked before release, therefore it is not clear if the same hornets kept coming to the artificial nest boxes to try in vain to catch a prey that they were apparently unable to kill. It seems then that besieging bumblebee nests (in the form of artificial nest boxes) would represent only a cost for hornets and once more I wonder if the visual resemblance with a honeybee hive, an ideal foraging situation for *V. velutina*, may not play a role in hornet attraction to these artificial nests.

Captured hornets were not marked before being released, as this was beyond the scope of the study and would have further disturbed their behaviour. However, it should be noted that sites presented many viable foraging opportunities other than the bumblebees themselves. As such, although hornets often failed to hunt bumblebees, they were still likely to successfully catch other insects, and exploit the available nectar resources. The term 'besieging' suggests large numbers of hornets clustering at the bumblebee colonies. This was not the case, and it should be noted that the majority of hornets visiting sites were able to exploit floral and prey resources other than the bumblebee colonies. Thus, while the attraction of hornets to bumblebee colonies may have imparted some costs in terms of time and energy, this is unlikely to have impacted the hornets at a colony level.

The authors themselves report that "hornets were preferentially attracted to colonies with a higher frequency of foraging traffic, and the duration of this attraction was significantly correlated with the probability of predation, suggesting that continuous prey flux is a requisite stimulus for the maintenance of predatory behaviour" (Discussion, lines 447-450). It is therefore possible that the

foraging traffic, made conspicuous by the entrance of the white nest boxes, might cause an experimental bias attracting hornets to an unrewarding artificial resource at an unnaturally realistic rate compared to wild bumblebee nests.

This is a reasonable point, however the foraging traffic at our colonies did not differ from the level of foraging traffic present in large wild colonies. We have also now included evidence of the threshold of foraging traffic at which hornets were attracted, which suggests that it is not just the physical entrance or colony design that is eliciting their interest. See figure: S4. Indeed, there is substantial evidence that hornets are attracted to any aggregation of foraging insects, despite these often being concealed within floral patches, or close to ground level (Williams 1988; Perrard et al. 2011). Further, as hornets are often attracted to superficially unrewarding prey resources (Williams 1988; Rojas-Nossa and Calviño-Cancela 2020; Rome et al. 2021), it is unlikely that the attributes of the nest boxes alone were responsible for their interest. We have consequently provided additional clarification of this in the discussion. See lines: 346-351.

Overall, I acknowledge the considerable effort and experimental work of the authors, but I feel that further information and clarifications concerning the actual occurrence and impact of interactions between *V. velutina* foragers and wild bumblebee colonies in the field are needed to corroborate and give the right credit to the results of the study.

We hope that we have answered these points by providing further clarification of the difference between using artificial nest boxes and surveying wild colonies. While it is not currently feasible to examine hornet behaviour at multiple wild bumblebee colonies of multiple species, we consider that our large and standardized experiment provides the best evidence thus far of interactions at colonies.

Referee: 2

Dear colleagues,

I reviewed the manuscript “Quantifying the Impact of an Invasive Hornet on Bumblebee Colonies” (O’Shea-Wheller et al.). This manuscript presents a very interesting work about the predation of *V. velutina* on Bumblebee colonies. Protocols are well done and the obtained results are very interesting to colleagues interested by the impacts of this invasive species. The manuscript is well written and easy to read.

I have some minor comments that the authors need to address to improve (a little) their manuscript.

Thank you for these positive comments.

Materials and methods:

- Line 102: Could the authors give to the readers more information about pathogens and parasites analyzed and about the methods used please?

We have now included further information along with a table detailing all pathogens and parasites screened for in the supplementary information. See table: S2.

Discussion:

- Line 455: I disagree with the authors, as Rome et al. (2011) showed that very few Bumblebees were predated by the hornet. They need to modify the sentence and to give the reference please.

We have modified this sentence as suggested, to reflect that our study is the first *colony-level* assessment of impacts, rather than simply the first assessment. See lines: 275-276.

- Lines 479-490: The discussion is interesting and the work needs further experiments to clarify the effects of the *velutina* presence in front of bee colonies. However, the stress of colonies by *velutina* could greatly impact bumblebees. More and more works demonstrate this effect on honeybees. I think it could be more or less the same in bumblebee colonies. Authors need to discuss this point. For example, there is the work of Leza et al (2019).

This is a relevant point, we have now included discussion of this possibility, along with the associated reference. See lines: 317-319.

- Line 521: A question, is it possible to estimate the threshold?

Yes, from our data it appears that hornets generally required mean departure rates of ≥ 20 bees per hour and mean return rates of ≥ 13 bees per hour to elicit sustained interest, as $>90\%$ of hornet presence occurred above this threshold. While such a threshold should be treated as relative to the specific conditions of our study, it does serve to suggest that sufficient activity is a prerequisite of hornet attraction. We have now included a figure detailing this along with resultant numbers of predation attempts in the supplementary information, and provided citations to support our discussion in the main text. See figure: S4, see lines: 215-216, 346-348, 352-354.

Referee: 3

This manuscript examines the important question of how an invasive hornet, *V. velutina nigrithorax*, impacts the fitness of a native bee species, the bumble bee, *Bombus terrestris*. The authors have taken great pains to replicate their experiment at multiple sites and to measure multiple variables, including bee health, bee fitness, bee and hornet behavior, and weather variables. The main findings are that 1) hornets tried to attack bees, but did not succeed; 2) in areas with more hornets, colonies weighed less, indicating lower fitness; and 3) hornet presence correlates with increased bee departures and returns. I think that the paper is well written (although the Discussion could be shortened, see below) and the data are correctly analyzed.

Thank you for these detailed comments.

I think that this data is certainly worthy of publication, but I think that the authors should more fully explore the possibility that some of these correlations are due to other factors, such as the landscape which could drive both hornet abundance and bee colony variables. For example, please discuss hornet abundance may not necessarily drive down colony weights. Perhaps there is another factor such as other prey items? *Bombus terrestris* does not comprise much of the hornet diet (see cited genetic analyses). If so, hornet abundance may be linked to their actual prey, including honey bee colonies. Adding this honey bee abundance to the models might be revealing.

This is a salient point, it is indeed possible that the correlations observed were due to additional unmeasured covariates, and we have now expanded upon this in the discussion. See lines: 306-309. Unfortunately, we were not able to comprehensively assess honey bee colony abundance in the vicinity of sites, as the majority of adjacent areas were private property with different owners, making surveys unfeasible within the time constraints of the study. Nevertheless, we have expanded upon the potential influence of this in terms of competition with the bumblebee colonies. See lines: 309-311.

I am not sure that it is correct to call the departures and returns at the nest entrance “foraging”. It is interesting that bee departures and returns were higher when there were hornets, so perhaps what is counted is not actually foraging but alarm behavior? This possibility should be mentioned.

We are confident that departures and returns did represent foraging, as both increased with the percentage of heterogeneous agriculture at sites, and peaked in mid-August. Further, bees generally returned to the colony with pollen, as would be expected for foragers. The relationship between such traffic and hornet presence is also more likely to be attributed to hornet attraction, rather than an alarm response, as colonies with high foraging traffic maintained this when hornets were not present. Furthermore, there was no apparent increase in traffic upon the onset of hornet presence or predation, rather, colonies that already had high foraging activity subsequently attracted hornets. We have now clarified this in the methods, results, and discussion. See lines: 183, 354-356, 504-505.

With respect to the colony setup (Fig. 1), is it possible that the tube used to monitor bee behavior also helped to prevent hornets from capturing bees? Such devices have been used to lower the success rate of *V. pensylvanica* entering and attacking honey bee nests. Attacks were also measured outside of the colony, but the potential effect of the tube is perhaps worth a mention.

We have now included some discussion to this end, clarifying that the tube itself is unlikely to have substantially altered predation success rates by hornets. From observations on honeybees, we know that *V. velutina* targets flying bees outside of the colony, and rarely enters to catch prey. Indeed, we observed the same trend with bumblebees in the present experiment—as hornets targeted bees flying from or to the entrance tube, rather than those in the tube itself. See lines: 367-370.

Finally, it would be good for the authors to clarify the expected link between pathogens and hornet presence. Is the idea that hornets will stress bee foragers and thereby increase their pathogen load? Since you have a limited sample size that is too small for a causal analysis, I suggest that this aspect of the paper be placed in the SI. This will also help reduce the length of the Discussion.

This is a reasonable point, we did not aim to assess links between pathogen prevalence and hornet presence, but rather undertook analyses to characterise any environmentally acquired infections. We have subsequently provided further clarification of this in the materials and methods section. See lines: 468-474. Additionally, we have moved the pathogen analyses into the supplementary information as suggested. See supplementary information lines: 104-136.

Other comments

L102 please list the pathogens and parasites that your screen could identify.

We have provided a table detailing this in the supplementary information. See table: S2.

L110 a map of the sites would be useful in the SI

Agreed, we have now added a figure detailing their location in relation to administrative municipalities, surrounding land cover, and satellite imagery. See figure: S1.

L113 please provide citation showing that VespaCatch can capture *V. velutina nigrithorax*.

We have now provided a suitable citation. See line: 407.

L452 Here you note that colonies were lighter in areas with more hornets and attribute this to foraging disruption. This supports the idea that bee departures and returns are not foraging but actually alarm.

This refers to the trend in colony weights over time rather than short-term changes due to foraging or alarm behaviour. Specifically, we found that colonies in areas with higher hornet levels appeared to show reduced growth in terms of weight gain. There are several possible explanations for this, with reduced foraging efficacy due to predation attempts by hornets being one of these. We have now provided further information detailing this prospective mechanism, along with clarifying our use of the term foraging traffic rather than alarm behaviour when describing bee departures and returns. Although it is possible that bumblebees experienced some level of alarm in response to hornets, this was not reflected in worker traffic changes when hornets were actively preying at colonies. Indeed, worker traffic responded only to land use and climatic conditions, as would be expected of foraging behaviour. See lines: 311-319, 504-505.

L546 Please expand upon how this negative correlation indicates that sublethal effects should be considered.

Results indicate that despite there being an apparent lack of direct mortality due to hornets attacking bees, there nevertheless is a negative correlation between hornet density and colony weight. This suggests that sub-lethal effects may play a role, and thus these should be considered. We have subsequently provided additional clarification of this point. See lines: 313-319, 389-390.

REVIEWERS' COMMENTS:

Reviewer #1 (Remarks to the Author):

The authors have satisfactorily responded to all my previous comments and I have no further requests.

Reviewer #3 (Remarks to the Author):

I believe that the authors have addressed my original questions and concerns, particularly about the potential effects of alarm behavior and the clear tube used at the entrance. I believe that this manuscript is now ready to be accepted.